# Peer review of "The Challenge for a Correct Diagnosis of Refractory Thrombocytopenia: ITP or MDS with Isolated Thrombocytopenia?"

_cancers, 2024, doi:10.3390/cancers16081462_

Round 1

Reviewer 1 Report

Comments and Suggestions for Authors

Kosmidou A. and coauthors in their paper “The challenge for a correct diagnosis of refractory thrombocytopenia: ITP or MDS with isolated thrombocytopenia?” review the modern concepts on the pathogenesis, diagnosis and treatment of immune thrombocytopenia (ITP) and the difficulties in differential diagnostics of the refractory forms of ITP and myelodysplastic syndrome (MDS) with isolated thrombocytopenia (MDS-IT). Although MDS-IT is a quite rare disease (no more than 10% of all cases of MDS) its misdiagnosis with the refractory ITP might lead to unappropriated and ineffective treatment of those patients. The authors analyze specific clinical characteristics and some diagnostic approaches that may help in distinguishing refractory ITP and MDS-IT. The review is well organized, well written and illustrated. However, there are some comments and suggestions for adding material that may be useful in addressing the main topic of the review.

Comments and suggestions

The authors do not pay much attention to the laboratory methods used for ITP differential diagnosis. Some of them are not included in the classic clinical guidelines for diagnosing this disease, but nevertheless might help to distinguish it from other variants of isolated thrombocytopenia. We are talking not only about antiplatelet autoantibodies, but also about the content of reticulated (immature) platelets, platelet size (mean volume and other indexes), the level of plasma thrombopoietin. These indicators are known to differ in patients with ITP and bone marrow hypoproductive thrombocytopenias. In the authors' opinion, can these methods be applied to the differential diagnosis of ITP and MDS?

Minor remark

Line 391. 5.1. Challenges in Diagnosis of ITP. Should be 5.1. Challenges in Diagnosis of MDS-IT

Author Response

Thank you for taking the time to review this manuscript, and for your recommendations. Please find the revised parts of the text highlighted in the re-submitted file. 

Talking about the diagnostic procedure of ITP, this part of the manuscript has been revised and extended to include some extra laboratory methods and their utility on the discrimination of ITP from MDS. Measurement of immature platelet fraction could serve for quantifying thrombopoietic activity and platelet production, but it would have limited clinical value for separating ITP from diagnoses with bone marrow failure syndrome, as immature platelets  have been shown to increase in number in patients with low platelet count independent of underlying diseases, and the pathophysiology of ITP includes -except for decreased platelet production- increased platelet destruction in peripheral blood, as well. Platelet size would definitely help diagnosing bone marrow syndromes, as it is only normal or increased in ITP patients. Measurement of TPO levels is not routinely used in the diagnostic procedure of ITP, however it could be proved helpful in confusing cases and for predicting response to treatment with TPO-RA. 

Reviewer 2 Report

Comments and Suggestions for Authors

Thank you for the opportunity to review the article ''The challenge for a correct diagnosis of refractory thrombocytopenia: ITP or MDS with isolated thrombocytopenia? “

This paper is comprehensive and well written. The subject is relevant and insufficiently covered in literature. There are minor corrections that are marked in the paper. Also, I recommend shortening the section on the etiology, diagnosis and treatment of ITP. Emphasize the part of the text related to the diagnostic approach of isolated thrombocytopenia.

I find this article worth publishing. 

Author Response

Thank you for your time reviewing this manuscript. According to the corrections you have recommended, we have revised the corresponding parts of the manuscript. Those can be found highlighted in the re-submitted file. Below, you can find point-by-point responses to your comments. 

Line 30-32: Comment 1: Make the sentence clear and shorter.

Response 1: The sentence has been made shorter. 

Line 34-37: Comment 2: Make the sentence clear and shorter. 

Response 2: The sentence has been modified to be clearer and shorter.  

Line 115: Comment 3: Put new references of epidimiological data, if available. 

Response 3: Epidemiological references have been updated, as recommended. 

Line 185: Comment 4: Put the new reference Provan 2019. 

Response 4: The reference with the number [25] has been updated to the new Provan 2019. 

Line 344: Comment 5: Please include the most recent reference N Vianelli 2022. 

Response 5: The recommended reference has been added. 

Reviewer 3 Report

Comments and Suggestions for Authors

The article entitled “The challenge for a correct diagnosis of refractory thrombocytopenia: ITP or MDS with isolated thrombocytopenia?” addresses an interesting clinical question regarding the corrected etiopathogenetic diagnosis, The authors exhaustively discuss the argument  exhaustively and the structure is well done. Anyway, I have some suggestiond to make. For example, in the section 3. Immune Thrombocytopenia I suggest to remove the term “….idiopathic thrombocytopenic purpura. Infact, the IWG defined the abbreviation in common use (ITP) to be Immune Thrombocytopenia (neither Idiopathic nor Purpura) (Lambert MP et al, Blood 2017). In addition in the section 3. Immune Thrombocytopenia the authors quote a concept “It is well established that bleeding events of ITP……. are not always related to severity of thrombocytopenia. This concept deserves to be addressed and I suggest adding  the reference (Middelburg RA et al, Hematology 2016). I think that this article is suitable for publication in a revised version.

Author Response

Thank you for taking the time to review this manuscript, and for your suggestions. Please find the revised parts of the text highlighted in the re-submitted file. 

The term 'idiopathic thrombocytopenic purpura' was used as a mention to the previously-used term, and that is why it has been used only once. Anyway, it has been removed, as the standardized terminology is now 'immune thrombocytopenia'. In addition, the sentence “It is well established that bleeding events of ITP……. are not always related to severity of thrombocytopenia" has been better illustrated and the suggested reference (Middelburg RA et al, Hematology 2016) has been added.